# Standardised 25-Step Traditional Thai Massage (TTM) Protocol for Treating Office Syndrome (OS)

**DOI:** 10.3390/ijerph20126159

**Published:** 2023-06-16

**Authors:** Wiraphong Sucharit, Neil Roberts, Wichai Eungpinichpong, Torkamol Hunsawong, Uraiwan Chatchawan

**Affiliations:** 1Research Center in Back, Neck, Other Joint Pain and Human Performance (BNOJPH), School of Physical Therapy, Faculty of Associated Medical Sciences (AMS), Khon Kaen University (KKU), Khon Kaen 40002, Thailand; wiraphongsucharit@gmail.com (W.S.); wiceun@gmail.com (W.E.); tkmhun@kku.ac.th (T.H.); 2Centre for Reproductive Health (CRH), School of Clinical Sciences, The Queen’s Medical Research Institute (QMRI), University of Edinburgh, Edinburgh EH16 4TJ, UK; neil.roberts@ed.ac.uk

**Keywords:** office syndrome (OS), myofascial trigger point (MTrP), pain, primary care, traditional Thai massage (TTM)

## Abstract

Traditional Thai massage (TTM) is a unique form of whole body massage practiced to promote health and well-being in Thailand since ancient times. The goal of the present study was to create a standardised TTM protocol to treat office syndrome (OS) diagnosed based on the identification of the palpation of at least one so-called myofascial trigger point (MTrP) in the upper trapezius muscle. The new 90 min TTM protocol, which was developed following appropriate review of the literature and in consultation with relevant experts, has 25 distinct steps (20 pressing steps, 2 artery occlusion steps, and 3 stretching steps). Eleven TTM therapists treated three patients each using the new 90 min TTM protocol. All of the therapists reported scores greater than 80% in respect to their satisfaction and confidence to deliver the protocol, and all of the patients gave the treatment a satisfaction score of greater than 80%. The treatment produced a significant reduction in pain intensity measured on a Visual Analogue Scale (VAS), with minimum and maximum values of 0 and 10 cm, of 2.33 cm (95% CI (1.76, 2.89 cm), *p* < 0.001) and significant increase in pain pressure threshold (PPT) of 0.37 kg/cm^2^ (95% CI (0.10, 0.64 kg/cm^2^), *p* < 0.05). The protocol was revised based on the feedback and the results obtained, and the new standardised TTM protocol will be applied in a randomised control trial (RCT) to compare the efficacy of TTM and conventional physical therapy (PT) for treating OS.

## 1. Introduction

Traditional Thai massage (TTM) is a unique form of whole-body massage that is a primary care method in Thailand for promoting physical health and mental well-being [1]. The client who receives the massage changes into a loose-fitting top and shorts and initially lies supine on a mat on the floor or on a low bed. The therapist will then apply rhythmic (see later description) manual acupressure along designated paths mapped on the surface of the body. In the Thai language, the paths, of which there are 10, are named Sen Seb lines [2]. Each Sen Seb line may extend along the length of one or more limbs and torso, and the massage is usually delivered along several Sen Seb lines in one portion of the body at a time. Following the Sen Seb lines ensures that the acupressure is applied only to parts of the body where the stimulus is safe (i.e., avoids pressure on major nerves and blood vessels and delicate regions, such as where the skull is thinnest) and where it will be most effective (e.g., in the belly of the muscle and progressing along the principal fibre direction) [3]. Typically, the massage begins at the left foot, progressing along each leg in turn, then on to the torso, and in the case of the arms, begins at the left shoulder and progresses to the hand along each arm in turn. From the supine position, the client may be turned first on their right side (thus presenting the left-hand side of their body for massage) and then on their left side or else proceed directly to prone lying, and in each case the massage is resumed proceeding from the feet towards the head [2]. The protocols may last one hour, 90 min, or often 2 h [4].

Pressing is the essence of TTM, and the rhythm of the acupressure at each point along the Sen Seb line has been described as comprising five distinct phases, applied as a compressive force orthogonal to the long axis of the muscle, namely, contact, push, pin, press, and remove contact [5,6]. This pulse, which depending upon the preference of the client or the desire to heighten the therapeutic effect, may be applied with soft, medium, or firm pressure of the thumbs, has two principal physiological effects, both of which are enhanced by the five-phase nature of the pulse and is generally a very pleasant experience, putting body and mind in a receptive, watching state. The mechanical pressure will stimulate an increase in the compliance of the blood vessels, which, in tandem with the massage of the tissues, will promote increased blood flow [7,8]. Furthermore, the pulse will have an effect on the nervous tissue and will be felt as both a local tactile experience and may also elicit a wider change in the central nervous system (CNS) [9].

Court-style TTM (CSTM), sometimes referred to as royal-style TTM, is a particular form of TTM that is typically applied in the treatment of a wide range of medical conditions [10,11,12,13], and the massage is usually given in a dedicated ward in a hospital or in a private clinic. In addition to delivering the acupressure pulse at evenly spaced points along the Sen Seb lines, when performing CSTM the therapist may pay particular attention to specific locations of interest, which are referred to as major signal points (MaSPs), and of which there are a total of approximately 150 located at well-defined anatomical locations throughout the body [7,8]. At MaSPs, it is posited that the acupressure pulse will have a greater physiological effect, especially increasing blood flow [7], and the duration of the pulse is typically increased from 10 s to 45 s [8]. However, in terms of the total duration, CSTM treatments may be relatively quick, with protocols typically lasting 30 to 45 min, and there are even treatment protocols that last between just 5 and 15 min, involving firm pressure being applied to only a small number of MaSPs, the exact constellation of which depends upon the condition being treated [4].

The most widely available form of TTM, sometimes referred to as folk-style TTM, is generally focused on promoting relaxation [1]. In contrast to CSTM, in TTM the therapist may often move the client into a series of what may be referred to as assisted yoga postures. This can allow the acupressure pulse to access and stimulate a greater proportion of the body and potentially also deeper tissues. Furthermore, pressure is frequently applied also via the palm and even the forearm or elbow, and for highly experienced clients the therapist may apply high levels of pressure to specific sites by stepping with due care and control onto the back, shoulders, and legs. There are also variants of TTM, such as oil massage, aroma massage, and Thai oil massage.

In TTM, typically at the very end of the protocol, the therapist may take the client through a routine comprising a number of passive stretches (i.e., the therapist initiates the movement, and the patient watches and allows the movement to occur) [14,15]. The client will generally sit up and, according to the flexibility of the client, the therapist will pick say 5 to 10 out of a set of approximately 40 well-established stretching manoeuvres. With the client remaining passive, the therapist will gently flex, extend, adduct, or abduct the arms and flex, extend, and rotate the whole torso and/or head and neck in the manner that is prescribed and within a safe range of movement that approaches the end point for the particular movement in the individual patient. Since the muscles have already been “warmed up” and the pain eased, there is an opportunity for the passive stretching “exercises” to improve the flexibility of the client [16].

CSTM and TTM both provide nourishment for the tissues, modulation of the nervous system, and differ in the respective emphasis that is placed in the case of CSTM on treating a clinical condition and in the case of TTM the general promotion of heath and well-being. The underlying principle of both approaches is that muscles respond positively to being pressed and stretched and which tends to occur naturally in the normal activities of daily living and may be enhanced by massage. In CSTM which is used for medical massage the emphasis is on pressing rather than stretching as the latter may be inappropriate to apply in patients who may be in pain or otherwise incapacitated. In TTM a combination of pressing and stretching is used and which is adjusted in terms of the position adopted and the level of force applied so as to be most appropriate for each client [4].

There is one more significant aspect of TTM protocols still to be discussed and which is frequently included in both CSTM and TTM protocols. This procedure is called “opening the wind gates”. In particular, beginning typically with the left femoral artery, and generally with the leg slightly adducted and flexed at the knee, a gentle pressure is applied on the surface of the body at a place where it is safe and convenient to start to restrict the blood flowing through the artery (i.e., in the groin) [17]. If, based on experienced tactile perceptions and observation, the TTM therapist judges it to be appropriate, the pressure is increased and held for up to 45 s. When the pressure is gradually released the blood, which is likened to a wind, will enter the leg with increased vigour and potentially access and nourish a greater volume of muscle than was occurring before the occlusion [17]. The procedure is repeated for the right femoral artery. Later in the massage a corresponding procedure will be performed for the left, and later the right, brachial artery in the upper portion of the underarm. In this case, the arm is extended straight out to the side along the same direction as the shoulder girdle and with the palm facing upwards.

Office work is typically sedentary with employees frequently having to adopt fixed positions for long periods of time and which has a potentially detrimental effect [18]. For example, musculoskeletal injury may occur. The potential combination of sensory, motor, and autonomic disturbance may be diagnosed as a form of myofascial pain syndrome (MPS), known as office syndrome (OS), which has a significant negative impact on daily living and the ability to work [19]. One way that has been used to diagnose OS is based on the identification of so-called myofascial trigger points (MTrPs) by palpation of taut bands in the muscle. MTrPs can be divided into two types, namely, active myofascial trigger points (ATrPs), which are a source of spontaneous pain (i.e., pain that occurs in the absence of any specific stimulus) and latent myofascial trigger points (LTrPs), which only give rise to pain when stimulated by, for example, compression. Both ATrPs and LTrPs arise because of prolonged work in static positions producing sustained muscle contraction and habitual muscle stiffness, in turn, leading to a reduction in blood circulation, ischemia, and hypoxia [20]. MTrPs are a major cause of pain and referred pain in office workers and make the fixed positions that the employee is required to adopt in the workplace highly uncomfortable and often lead to absence from work [18,21,22]. From the descriptions already given, it may be apparent that TTM is ideally suited to the treatment of OS.

There are several previous reports of both CSTM and TTM having been applied to treat OS [4,23,24]. Most published research has used CSTM protocols, which have been reported to reduce pain intensity, increase the pain pressure threshold (PPT), increase range of motion and body flexibility, and improve quality of life [10,13,25]. However, there are also published studies in which TTM has been reported to have no beneficial effects in the treatment of OS [26]. The disparity in results is potentially because of different criteria having been used to define OS, use of different TTM protocols, and variations in the skill and experience of the TTM therapists. The goal of the present study is therefore to develop a safe, effective standardised TTM protocol for treating OS that can be potentially used in randomised control trials (RCT’s). The starting point for the development of the new 90 min TTM protocol was the protocol used in a previous study by Bennett et al. (2016) [1]. This starting protocol was divided into 25 steps, refined, and organised in consultation with experts to produce a protocol that was considered to be the most effective for treating OS. Subsequently, 11 TTM therapists, each with at least 2 years of professional working experience, attended a week-long training course to learn how to administer the new TTM protocol, which they each subsequently applied to treat three patients with OS. Pre- and post-treatment pain intensity and pain pressure threshold (PPT) were measured, and the satisfaction and confidence of the therapists to deliver each step of the TTM protocol were recorded together with the overall satisfaction scores reported by the patients. All findings were reviewed, and several adjustments were made to the protocol. The resulting standardised 25-step TTM protocol provides guidelines for the treatment of OS, a protocol for use in RCTs, and a potential benchmark for validating the scientific basis of TTM.

## 2. Materials and Methods

### 2.1. Design of the 25-Step Protocol

The study was approved by the Research Ethics Committee of Khon Kaen University (KKU), Khon Kaen, Thailand (Ref: HE631017), and the work was carried out in the Department of Physical Therapy, Faculty of Associated Medical Sciences (AMS) at KKU.

The new 25-step TTM protocol is similar to that used in a previous study [1] and includes specific developments for the treatment of patients with OS. The acupressure technique employed for the pressing steps has been described in previous studies [5,27], and the protocol is administered to patients in a comfortable setting with the room temperature controlled between 25 and 28 °C.

The subdivision of the TTM protocol into 25 steps is helpful for three reasons. Firstly, at the design stage, it is convenient to be able to discuss individual components of the protocol. Secondly, at the stage of a trial experiment it is convenient to be able to obtain evidence regarding the satisfaction and confidence of the TTM therapists and to adapt and revise appropriate steps in the protocol. Thirdly, once the final protocol is confirmed, it is more convenient to teach to new therapists and also provide a helpful structure for self-learning.

The 25 steps of the protocol are of three types, namely, 20 pressing steps, 2 artery occlusion steps, and 3 stretching steps, and the protocol is administered in 5 stages in which the patient lies supine, then on their side (right then left), then prone, and then supine again. The rationale for the total of 20 pressing steps is that 20 is reasonable for the number of different mutual positions that the patient and therapist can conveniently adopt in order for the therapist to be able to readily press at approximately 5 to 10 cm intervals along all of the chosen Sen Seb lines and which in the present protocol is reduced to 8 rather than 10 because line 9 (Sukhumung) and line 10 (Sikinee), which relate to the reproductive organs, were omitted.

The original version of the 25-step TTM protocol (see left-hand column of Table 1) has 8 pressing steps applied with the patient in a supine position, 8 with the patient in a side lying position, 3 with the patient in a prone position, and one after the patient resumes the supine position. The 2 artery occlusion steps are applied during the initial supine lying. Two of the stretching steps occur in the side lying position and one in the final supine position.

Three of the 20 pressing steps in the original protocol (left-hand column of Table 1) correspond to major signal points (MaSPs) (see Figure 1), namely, Step 5 for the palm of the hand (MaSP-Pal-3 and MaSP-4), Step 9 for the pectoralis (MaSP-SH-4), and Step 14 for the gluteus (MaSP-LL1, MaSP-LL-2, and MaSP-LL3). These MaSPs are all relevant for the treatment of OS because of the way in which an office worker will frequently be seated and how they may be holding their shoulders and hands to perform tasks such as typing and writing.

In general, the TTM therapist will take on the order of 10 s to perform each one of the presses (which may range in duration from 10 to 30 s) in the 17 pressing steps that do not correspond to MaSPs. In the case of the MaSPs, the pressing time is increased to up to 45 s.

With regard to the stretches, these are illustrated in the Figure 2. The decision to include the first two stretches was made based on standard practice in physical therapy for treating OS. They were performed one after the other in the relevant stages of the protocol when the patient first lays on their right side and then on their left side. The first stretch (Step 19) is more precisely described as the mobilisation of the scapula (see Figure 2b), and the second stretch (Step 20) follows with the aim of extending the lower body to connect with the potential release of the shoulder produced by the mobilisation of the scapula. The third stretch (Step 25) is the rotation of the trunk first to the right side and next to the left side while the patient lies supine in the final phase of the protocol. This stretch is commonly performed in TTM practice, although not usually at the very end of the massage.

### 2.2. Investigation of the Efficacy of the 25-Step Protocol

An experiment was carried out to test whether the 25-step TTM protocol that was developed for treating patients with OS was fit for the purpose and suitable for use in an RCT. The goals were to ensure that the protocol could be readily learnt to the required proficiency by a group of TTM therapists so that the therapists felt satisfied with the training and confident to apply their new skills. In addition, it was checked that a cohort of patients with OS in whom the newly trained therapists apply the TTM protocol felt positive about the experience and that there was indication that the treatment is effective.

### 2.3. Recruitment of Therapists and Patients and Treatment with TTM

A group of TTM therapists were recruited via notices posted on bulletin boards and via oral requests made at private TTM clinics, hospitals, and other public service departments in Khon Kaen Province between September and October 2020. Interested participants were provided with an explanation of the purpose of the study and a description of the procedures that would be followed. Inclusion criteria were female or male TTM therapists aged between 30 and 60 years who had obtained a certificate to confirm attendance for 150 h of training at a course organised by the Department of Thai Traditional and Alternative Medicine, Ministry of Public Health, and who had obtained experience in working as a TTM therapist for at least 2 years. The number of therapists to be recruited was decided according to the aim that 80% of them would self-report a satisfaction score of >80%. The method described by Chow (2003) and Ngamjarus Chongsuvivatwong (2014) [28,29] was applied with a significance level set to α = 0.05 and power to 80%, and a clinically meaningful difference in satisfaction was taken to be 80% − 45% = 35%.

The cohort of office workers was recruited using a similar invitation process to that which had been used to recruit the TTM therapists. The inclusion criteria were female or male office workers aged between 20 and 60 years and whom W. S., who is a qualified physical therapist with more than eight years of experience working in the field of musculoskeletal physical therapy, diagnosed as having MPS based on the identification by palpation of an MTrP in the upper trapezius muscle. The exclusion criteria were history of any disease or other disorders that could cause potential confounding factors and contraindication of receiving TTM. A power calculation was performed to calculate the number of patients to be recruited based on data that were published by Buttagat et al. (2012) [15]. These authors reported that a pre-treatment pain intensity, recorded on a Visual Analogue Scale (VAS) with minimum and maximum values of 0 and 10 cm, was 5.94 (SD = 1.58) reduced to 3.88 (SD = 2.39) following a single TTM massage, which corresponds to a mean difference among VAS pain intensity scores of 2.06. The data indicate that a total of 33 patients needed to be recruited for the present study in order for a significant difference in pain intensity to be detected using a paired sample *t*-test.

The TTM therapists attended a 5-day training course, which took place at AMS at KKU to learn how to deliver the new 90 min 25-step TTM protocol. Training was provided by W. S. and two qualified TTM therapist assistants. For the first three days, the TTM therapists acquired essential knowledge concerning the new TTM protocol and practiced the 25 steps working on each other. Subsequently, each of the 11 TTM therapists was randomly assigned to deliver the 90 min 25-step TTM protocol to treat 3 of the office workers. The treatments were delivered on days 4 and 5 of the training course and prior to the TTM massage, the health status and vital signs, including blood pressure, of each patient with OS were checked.

### 2.4. Outcome Measures

The following outcomes were recorded:For each of the 25 steps, each TTM therapist provided responses on a 3-point Likert scale regarding whether they considered that they had learned the specific action to be performed in terms of (a) the Sen Seb line to be followed, (b) the hand and body position to adopt, and (c) the force to apply;For each of the 25 steps, each TTM therapist provided responses on a 3-point Likert scale regarding whether they felt confident to be able to deliver the treatment in terms of (a) the Sen Seb line to be followed, (b) the hand and body position to adopt, and (c) the force to apply;Each TTM therapist was asked if there were any recommendations that they would like to make;Each OS patient provided responses on Likert’s 3-point scale for 9 items regarding whether they were satisfied with the TTM treatment they received;For each OS patient, prior to treatment and again immediately following treatment several outcome measurements were recorded. These were the intensity of the pain which they were currently experiencing as measured using a numerical VAS scale, ranging from 0 to 10 cm, with 0 corresponding to no pain anywhere and 10 corresponding to the worst pain imaginable, and PPT was measured using a pressure algometry (OE-220, ITO Co., Ltd., Tokyo, Japan) for the central part of upper trapezius muscle on the most painful side. The intra-rater repeatability and inter-rater reproducibility of the PPT measurements were tested in 28 volunteers before recruiting the participants. The results showed that the ICC was 0.94 (*p* < 0.001) and 0.97 (*p* < 0.001), respectively;Each OS patient was asked to report any side effects, such as slight bruising, mild headache, tiredness, increased discomfort or soreness, that were present immediately after receiving treatment and one day later;Each OS patient was asked if there were any recommendations that they would like to make.

### 2.5. Statistical Analysis

Data analysis was performed using IBM SPSS version 28 (Copyright KKU, Thailand). The baseline demographic information, including descriptive statistical analysis, is presented for the 11 TTM therapists and 33 office workers. For the continuous data, the mean values ± SDs are presented, and for the categorical data, the percentages are presented. The normality of the outcome measures was assessed using the Shapiro–Wilk test. The mean pre- and post-treatment values of the pain intensity and PPT are reported with 95% confidence intervals (CIs), and a paired t-test was used to test whether the difference is significant. R Statistical Software (v4. 1.2; R Core Team 2021) was used for the preparation of the figures. A preliminary analysis of the results of the experiment performed to investigate the efficacy of the 25-step TTM protocol was published by Sucharit et al. (2022) [3].

## 3. Results

### 3.1. Participants

Eleven TTM therapists, recruited from eight different private (72.7%) and three government (27.3%) clinics, provided fully informed written consent of their willingness to participate. The mean number of hours of training completed by the TTM practitioners was 202.9 ± 91.3 h, and the mean number of years of working experience was 4.1 ± 1.1 years in which they massaged an average of 4.8 ± 0.8 clients per day.

A total of 33 patients with OS were recruited and all gave fully informed written consent regarding their willingness to participate. Their mean age was 36.5 ± 10.5 years, and their mean BMI was 22.42 ± 3.51 kg/m^2^. Nineteen were female and 14 were male. Most worked as private employees and twenty-three (69.7%) were computer users. More than 60% of the patients received regular pain management, and the MTrP used to diagnose MPS was located in the right upper trapezius in 24 (72.7%) of the participants and in the left upper trapezius in the remaining participants. Among the patients with OS, seven (21.1%) reported having underlying disease not requiring their exclusion from the study (e.g., allergy, stage 1 hypertension), an average number of hours worked per day of 7.85 ± 2.34 h, and an average number of days worked per week of 5.45 ± 0.87 days. The eighteen (54.4%) patients with OS had worked in their present occupation for an average of 12.21 ± 10.03 years (range: 1 to 39 years). According to self-report using a VAS, the pain intensity was reported as severe by 9 (27.3%) patients, as moderate by 18 (54.5%) patients, and as mild by 6 (18.2%) patients.

#### 3.1.1. Therapists

The scores of the 11 TTM therapists with respect to whether they felt satisfied in having learnt and confident to follow the Sen Seb lines (i.e., line), adopt an appropriate hand position (i.e., hand), and apply an appropriate force (i.e., force) at each step are displayed in Figure 3. In each panel of Figure 3, the satisfaction scores are plotted on the left and the confidence scores on the right; a red symbol denotes satisfied or confident, a blue symbol denotes not satisfied or not confident, and an open circle denotes that the therapist is undecided. In Figure 4, the data presented in Figure 3 have been averaged across all 11 TTM therapists. The confidence and satisfaction scores that were recorded for the individual TTM therapists and plotted in Figure 3 were averaged across all 11 therapists, and the results are shown in Figure 4, where the average confidence scores are plotted above the average satisfaction scores. For each of line, hand, and force, the symbol is coloured red when the majority of the therapists gave a positive score, an open circle when an equal number of therapists gave positive and negative scores, and coloured blue when the majority of the therapists gave a negative score. The large number of red symbols in both Figure 4 and Figure 5 indicates that the TTM therapists were generally satisfied that they knew what to do and were confident of being able to carry out the steps of the massage protocol correctly. The main concerns were related to the first two stretches, which are of the upper body (step 19) and lower body (step 20) and performed with the patient in the side lying position. There was also some concern regarding the hand position to adopt in pressing of the lateral leg and thigh (Step 13), as well as the amount of force to apply in the pressing of the anterior leg in the supine lying (Step 2), pressing of the gluteus MaSP also in the side lying (Step 14), and pressing of the posterior leg in the prone (Step 21) positions.

#### 3.1.2. Outcome Measures Recorded for Patients

The pre-treatment pain intensity was 5.15 ± 1.97 cm and reduced to 2.82 ± 2.51 cm after treatment. The decrease of 2.33 cm (95% CI (2.89 to 1.76 cm), *p* < 0.001) is statistically significant. The pre-treatment PPT was 1.83 ± 0.74 kg/cm^2^ and increased to 2.20 ± 1.05 kg/cm^2^ after treatment. The increase of 0.37 kg/cm^2^ (95% CI (0.10 to 0.64 kg/cm^2^), *p* < 0.05) was statistically significant. The office workers reported an average score of more than 80% on the Likert scale for their satisfaction with the massage treatment.

A summary of the responses of the 33 office workers to receiving the TTM protocol is presented in Figure 5. The left-hand portion of Figure 5 refers to those patients in whom the massage produced a reduction in pain intensity from either severe or moderate to mild and, hence, can be considered successful. The middle portion of Figure 5 refers to those patients in whom the massage produced a reduction in pain intensity from severe to moderate and, hence, can be considered beneficial. The right-hand portion of Figure 5 refers to those patients in whom the massage did not produce any change in the pain intensity score. The change in the pain intensity is denoted by the length of the downward pointing black arrows, and the change in the PPT is denoted by the length of the upward pointing red arrows.

In Figure 6, the satisfaction expressed by each patient in the massage they received is plotted against the difference between the pre- and post-treatment pain intensity scores. In the left-hand panel, the data are presented for all of the individual office workers, and in the right-hand panel the data for the three OS patients treated by each therapist are averaged. Inspection of the two panels reveals that the massage generally produced a reduction in the pain intensity, the extent of which was not closely related to the satisfaction expressed by the patient. On average, the satisfaction expressed by the patients was similar and the pain relief they received was similar irrespective of which TTM therapist performed the massage.

No severe adverse events were reported. However, two TTM therapists (18.2%) reported experiencing some pain in the thumbs, hands, and upper back. In the case of the office workers, one (3.0%) reported feeling uncomfortable immediately after receiving treatment, and three (9.1%) reported mild muscle soreness, which was relieved in one participant by taking paracetamol. At two-days follow-up, no office workers reported any persisting side effects.

## 4. Discussion

The goal of this study was to develop a standardised TTM protocol for the treatment of OS that is straightforward for TTM therapists to learn to perform and which is well accepted by patients and is effective. Based on the reports completed by the TTM therapists with regard to their satisfaction in learning the protocol and their confidence in being able to apply the protocol, three notable revisions were made to the protocol. Firstly, and most notably, the stretching component of the protocol was revised so that the two stretches that had been proposed to be incorporated based on physical therapy practice (shoulder mobilisation and lower body extension) were found to be challenging for the TTM therapists to perform and were removed and replaced by more usual upper body and lower body stretches used in TTM practice. Furthermore, all three stretches were moved to the end of the protocol as is customary in TTM and performed in the order from lower body to torso to upper body (step 23, step 24, and step 25) which is, again, usual practice in TTM, and with the numbering of the intervening steps revised accordingly (see Figure 7). 

The next revisions to be described all occur before the first stretching step. In particular, so that the therapist may be more easily able to apply sufficient force, instead of the thumb, the heel of the hand is used for pressing on the anterior leg and thirdly (step 2), the position of the thumb is changed to the side thumb for pressing of the gluteus MaSP (step 14) and the position of the heel of the hand is changed to side palm press for pressing on posterior leg (Step 21). Furthermore, to ensure convenient access, the number of Sen Seb lines to be followed in the pressing of the lateral leg and thigh is reduced from three to two (step 13).

The 90 min 25-step standardised TTM protocol that has been developed for treating OS will be taken forward for use in an RCT to compare the performance of TTM and conventional physical therapy for treating OS. A detailed description of the new standardised 25-step TTM protocol can be found in the Appendix A, where Appendix A contains information on the pressing techniques, and Appendix A contains a full description of each of the 25 steps. It is hoped that these resources will allow others to readily learn and use the TTM protocol in their research and potentially adapt the TTM protocol for application in studies of other disorders and syndromes. The development of standardised TTM protocols is an important foundation for strengthening the scientific basis of the application of TTM.

TTM is a hands-on procedure, and the most important principle underlying all training courses is to ensure the safety of both therapist and patient. As has been discussed, the benefits may be increased if the pressure of the acupressure is increased from soft to moderate and even firm, but this is done in consultation with the patient and always with the flexibility to change during the course of massage. The map of the Sen Seb lines guides the therapist with regard to where on the body the massage should be focused to avoid injury from inappropriate force being applied to nerves, blood vessels, and joints. In addition, the therapist is taught how to perform the assisted movements, wind gate procedures, and passive stretching without injury and when to omit.

There is strong evidence that MTrPs accumulate in the muscles of the neck, shoulders, and back [30,31,32], which are eccentrically contracted for long periods of time in the fixed positions that office workers frequently adopt. If the pressing in any of the 20 pressing steps happens to coincide with an ATrPs, then this is likely to immediately increase the pain intensity, as it is in these places that pain sensitizing substances may have accumulated and this could influence whether PPT is increased or lowered following the massage (see below) [33,34]. Following the trajectory of the Sen Seb lines will make sure that the rhythmic acupressure massage reaches the majority of ATrPs and LTrPs [30].

The application of the new 25-step 90 min TTM protocol produced a reduction in the pain intensity in 32 of the 33 patients with OS. For 16 of the patients, the pain intensity was reduced from severe or moderate pain to mild pain, and in all these cases the PPT was found to increase. However, for four of the nine patients with severe pain, the PPT was observed to be decreased immediately after massage, which may indicate that patients with severe pain should not be treated with TTM, as they may exhibit possible adverse effects. The relatively small number of OS patients that were studied means that it was not possible to perform subgroup analyses to investigate the potential effects of age and sex, and this is an interesting topic for future research.

A possible mechanism that explains the development of OS is that a combination of postural and psychological stress causes the stiffness of UT muscle to increase, leading to functional limitations, impaired blood circulation, and accumulation of pain-sensitizing substances [34], especially in MTrPs [35]. In OS patients, MTrPs are commonly found to lie along the length of the main axis of the antigravity muscles [32]. The 25-step TTM protocol was designed according to the concept of delivering a treatment that is of benefit to the whole body, consisting of 20 steps of acupressure massage, 2 occlusion steps, and 3 steps of passive stretching. Several previous studies have indicated that acupressure massage [36,37], occlusion [17], and passive stretching [38] can all promote an increase in blood circulation, which may reduce the concentration of pain-sensitizing substances in stiff muscles [34,39] and result in a reduction in pain intensity and increase in PPT. Other possible mechanisms for the beneficial effect of TTM have been discussed by Keeratitanont et al. (2015) [16].

No severe adverse events were reported by either the TTM therapists or the OS patients. Four patients did, however, indicate that they experienced minor side effects, and one patient took paracetamol. In all four patients, the symptoms subsided within two days, and receiving the massage was always reported to have been a positive experience overall. These findings are consistent with a previous study in which mild aching in the shoulders was reported by participants who had never previously received TTM [11]. There were also no severe adverse events reported in studies of other massage therapies, such as Swedish massage, deep tissue massage, and trigger point therapy [40].

Two TTM therapists reported minor side effects and, in particular, relatively mild pain affecting the thumbs. In a study by Bennet et al. (2016) [1] and in the present study, the TTM therapists were encouraged to apply the pressing in the manner of CSTM. Thus, they used the digits of the hand, whereas TTM is generally practiced in which pressing may additionally be applied via the palm and even the forearm or elbow. Therapists who routinely deliver CSTM massage may perform finger strengthening exercises and be taught how to refine their practice such that, for example, rather than depending solely on the fingers, pressure can be delivered via appropriate connection with the torso “through” the fingers. The TTM therapists who were recruited for the present study were not familiar with using the pressing technique of CSTM, and there was not sufficient time to become entirely proficient during the short training course. Indeed, after being trained in TTM and practicing for several years, it is likely that a TTM therapist would find it somewhat challenging to change to being a CSTM therapist.

After participating in the 5-day training course, all 11 TTM therapists were certified as being competent to deliver the standardised TTM protocol. The effectiveness of the protocol might be further enhanced by a refresher training course. In particular, whilst the TTM therapists had no difficulty following the Sen Seb Line, they are likely to gain from reviewing the recommended positioning between therapist and patient and the level of force to be applied.

To the best of our knowledge, this is the first time in which a standardised TTM protocol has been developed supported by, on the one hand, a training programme to ensure the satisfaction and confidence of therapists to perform the prescribed massage and, on the other hand, with evidence that the treatment is well received by patients and shows the promise of being effective prior to embarking on an RCT. The approach is consistent with ensuring the standards of excellence in the conduct of scientific research and, especially, progress to an RCT, as well as in the safe and effective delivery of health interventions. In the case of the RCT, the 90 min 25-step standardised TTM protocol will be applied to treat each patient three times per week for two weeks.

## 5. Conclusions

A 90 min 25-step standardised TTM protocol was developed that is easy to learn and practice and which been shown to be well received and effective for treating patients with OS who have confirmed MPS involving the presence of at least one MTrP. The protocol is to be applied in an RCT to compare TTM and conventional physical therapy in the treatment of OS, and it is a resource that can potentially be adapted by other researchers interested in investigating the scientific basis of TTM.

## Figures and Tables

**Figure 1 ijerph-20-06159-f001:**
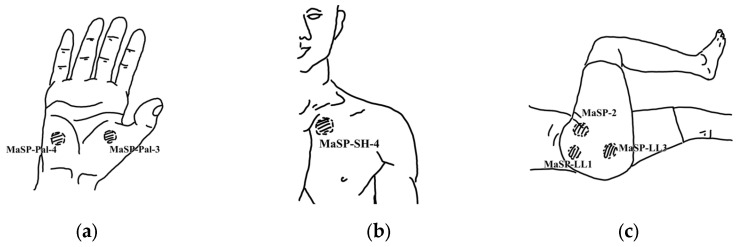
Three of the twenty pressing steps are major signal points (MaSP): (**a**) MaSP of wrist/palmar at signal 3 and 4; (**b**) MaSP of shoulder at signal 4; (**c**) MaSP of lateral side of leg at signal 1, 2, and 3.

**Figure 2 ijerph-20-06159-f002:**
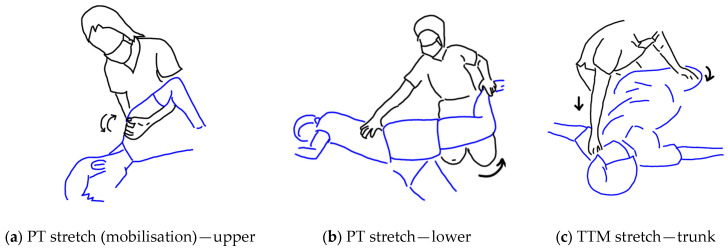
Stretching steps were designed based on standard practice in physical therapy: (**a**) shoulder and scapula stretching; (**b**) hip flexor and thigh stretching; (**c**) trunk stretching.

**Figure 3 ijerph-20-06159-f003:**
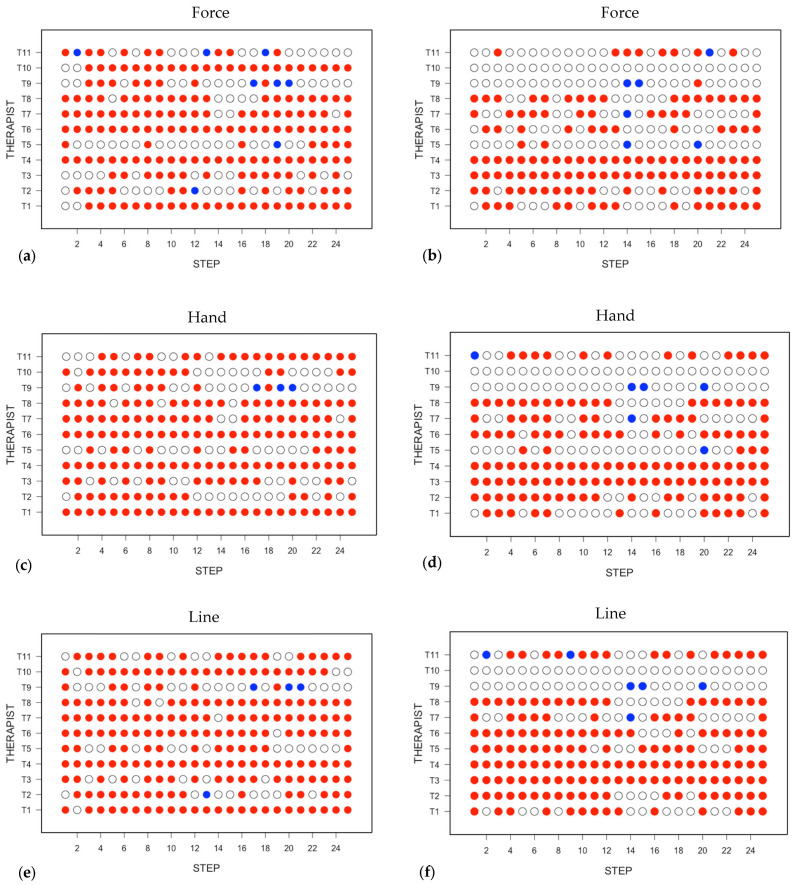
The responses from the 11 TTM therapists with regard to whether they felt satisfied that they understood the procedures to follow in each step in the TTM protocol and confident to administer the massage at each step. In each panel, the satisfaction scores are plotted on the left and the confidence scores on the right: (**a**,**b**) what force to apply; (**c**,**d**) what position to place the hands; (**e**,**f**) following the Sen Seb line. A red symbol denotes satisfied or confident, a blue symbol denotes not satisfied or not confident, and an open circle denotes that the therapist is undecided.

**Figure 4 ijerph-20-06159-f004:**
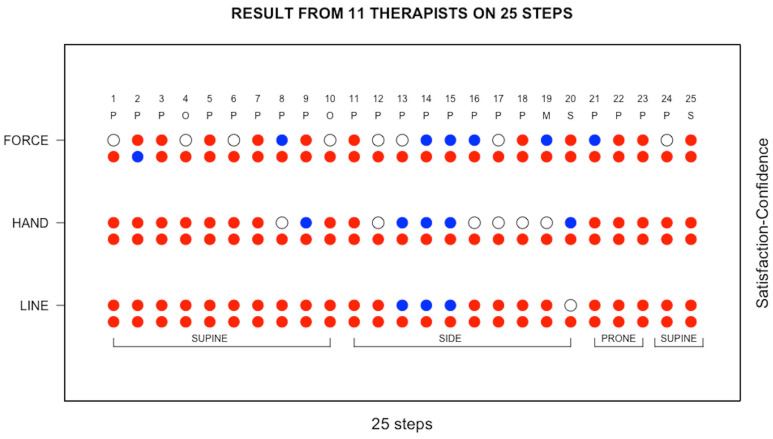
The data presented in Figure 3 were averaged across all 11 therapists, with the average confidence scores plotted above the average satisfaction scores. A red symbol denotes satisfied or confident, an open circle denotes undecided, and a blue symbol denotes not satisfied or not confident. P, pressing; O, artery occlusion; M, mobilisation; S, stretching.

**Figure 5 ijerph-20-06159-f005:**
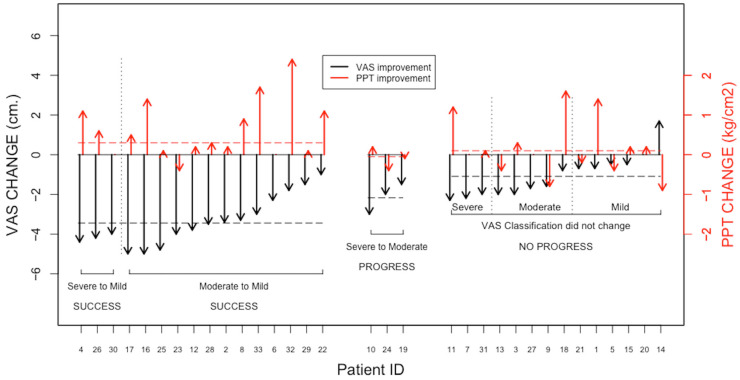
Summary of the changes in the VAS and PPT measured for the 33 patients with OS as a result of receiving the TTM treatment. Success: massage produced a reduction in the pain intensity from either severe or moderate to mild; progress: massage produced a reduction in the pain intensity from severe to moderate; no progress: massage did not produce any change in the pain intensity score. Black arrows denote the change in pain intensity, and red arrows denote the change in PPT.

**Figure 6 ijerph-20-06159-f006:**
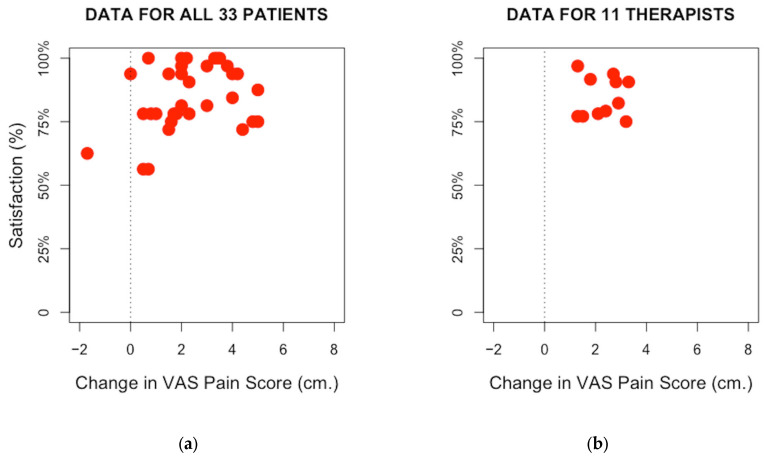
Satisfaction expressed by each patient in respect to the massage they received plotted against the difference between the pre- and post-treatment pain intensity scores, represented by the red spot: The vertical dotted line indicates the differentiation between pain improvement and worsening; (**a**) all individual office workers; (**b**) average of three OS patients treated by each therapist.

**Figure 7 ijerph-20-06159-f007:**
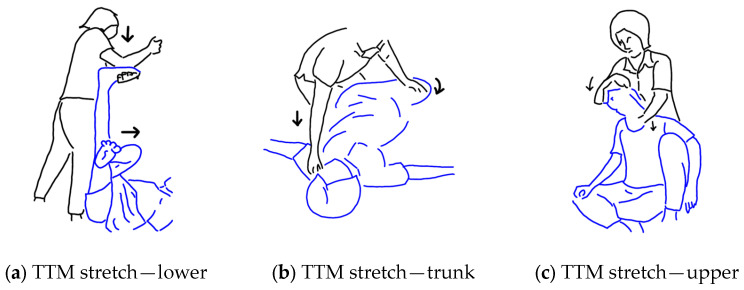
Stretches was made based on TTM practice: (**a**) lower extremity stretching; (**b**) trunk stretching; (**c**) upper trapezius stretching.

**Table 1 ijerph-20-06159-t001:** The 25 steps of the protocol: press, pressing steps; PT stretch, stretching steps of physical therapy; TTM stretch, stretching steps of traditional Thai massage.

Step	Original	Revised	Number of Pressing Points	Position of Patient
1	Press—Dor. Foot	Press—Dor. Foot	9	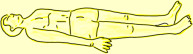
2	Press—Ant. Leg and Thigh	Press—Ant. Leg and Thigh	9
3	Press—Lat. Thigh	Press—Lat. Thigh	6
4	Occlusion—Femoral	Occlusion—Femoral	1
5	Press—Pal. Hand ***	Press—Pal. Hand ***	2
6	Press—Ant. Arm	Press—Ant. Arm	9
7	Press—Dor. Hand	Press—Dor. Hand	6
8	Press—Post. Arm	Press—Post. Arm	9
9	Press—Ant. Shoulder ***	Press—Ant. Shoulder ***	1
10	Occlusion—Brachial	Occlusion—Brachial	1
11	Press—Plant. Foot	Press—Plant. Foot	9	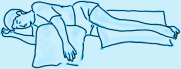
12	Press—Med. Leg/Thigh	Press—Med. Leg/Thigh	9
13	Press—Lat. Leg/Thigh	Press—Lat. Leg/Thigh	12
14	Press—Gluteus ***	Press—Gluteus ***	3
15	Press—Lat. Lower/Upper Back	Press—Lat. Lower/Upper Back	30
16	Press—Scapular	Press—Scapular	12
17	Press—Post. Shoulder/Neck	Press—Post. Shoulder/Neck	12
18	Press—Lat. Shoulder/Head	Press—Lat. Shoulder/Head	12
19	PT Stretch—Upper	Press—Post. Leg	6	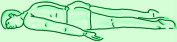
20	PT Stretch—Lower	Press—Post. Thigh	6
21	Press—Post. Leg	Press—Post. Lower/Upper Back	24
22	Press—Post. Thigh	Press—Head/Face	12	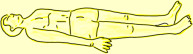
23	Press—Post. Lower/Upper Back	TTM Stretch—Lower	3
24	Press—Head/Face	TTM Stretch—Trunk	3
25	TTM Stretch—Trunk	TTM Stretch—Upper	4	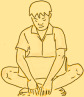

Red line, occlusion steps; Blue line, stretching steps; ***, Denotes major signal points; Dor., dorsal; Plant., plantar; Pal., palmar; Ant., anterior; Post., posterior; Med., medial; Lat., lateral.

## Data Availability

Not applicable.

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
