# Peer review of "Standardised 25-Step Traditional Thai Massage (TTM) Protocol for Treating Office Syndrome (OS)"

_ijerph, 2023, doi:10.3390/ijerph20126159_

Round 1

Reviewer 1 Report

Thank you for an opportunity to review this interesting manuscript. In general, it is well written. However, the manuscript needs some revisions for a publication.

Abstract

Line 15-18. Does that mean that one therapist treated each patient for 90 minutes? or one therapist treated three patients in 90 minutes? In the latter case, how long did the treatment take per patient? Please be accurate in Methods.

Line 21, 398. What does the values (2.89 to 1.76) mean? And I suggest that you provide the units.

Line 21, 22 398, 400. If the values (2.33 and 0.37) are an average of 33 patients, please also show the SD.

Line 22, 400. What does the values (0.10 to 0.64 kg/cm2)mean?

Introduction

Line 56-60. I suggest that you insert citations.

Line 67-69. The meaning of the sentence is unclear, especially what does “feeling more alive” mean? I suggest that you explain that in a little more detail and/or insert citations.

Line 70-72. I suggest that you insert citations.

Line 81. Specifically I want you to explain what “physiological effect” is.

Line 195. Is the sentence “11 TTM therapists, each with over 5 years of professional working experience, …” true? On line 358, you described “the mean number of years of working experience was 4.1 ± 1.1 years …”. This inconsistency can confuse the reader.

Materials and Methods

Line 333-337. When did you measure pain intensity? Immediately after the treatment? Also, in which body part did you measure pain intensity? Please be accurate.

Results

Line 363-365. I suggest that you provide the definition of “patients with OS” in this study and describe the symptoms of OS that the patients had in as much detail as possible. Based on your explanation,

Line 374. It is unclear how the average was calculated and how the plots in figure 5 were decided.

Line 398-400. How did you measure pressure pain threshold? What kind of equipment was used in the measurement? Please be accurate in methods.

Line 404, 406. I suggest that you specify whether a, b, or c in Figure 6.

Figure 4 and 5. “not satisfied or not confident” was denoted by a blue symbol in Figure 4, whereas it was denoted by a green symbol in Figure 5. Unless there is a special reason, please unify the colors in the two figures for the avoidance of confusion.

Figure 6. The sentence on line 397-400 and (a) and (b) of Figure 6 has the almost same content. I suggest that you reconsider whether you should include figure (a) and (b).

Figure 6. The meaning of arrows in Figure (a) and (b) is unclear. The meaning of the color difference (black, blue, and red) is also unclear. Please be accurate.

Discussion

Line 438. I suggest that you interpret the finding of this study in a little more detail. For example, what are the possible mechanisms of significant improvement in the average values of PPT and VAS? Also, what is the possible reason that it was effective for some patients but not for others? This study included patients of different ages/genders. Is there any possibility that the treatment effects varied depending on ages/genders? In addition, based on findings of this study, how can the new TMM be applied practically to clinical situations?

Author Response

Standardised 25 Step Traditional Thai Massage (TTM) Protocol for Treating Office Syndrome (OS) (ijerph-2288455)

We are very grateful to the two Reviewers and Editor for the very helpful reports they have kindly provided on the above manuscript that we have submitted for consideration for publication in the journal International Journal of Environmental Research and Public Health (IJERPH). The feedback is encouraging and we are pleased that the Reviewers consider the results that are presented to be clinically important. We have responded in full to all of the feedback that has been provided as is described below. All page numbers refers to the manuscript file with tracked changes.

Response to Reviewer #1

Abstract

1. Comment

Line 15-18. Does that mean that one therapist treated each patient for 90 minutes? or one therapist treated three patients in 90 minutes? In the latter case, how long did the treatment take per patient? Please be accurate in Methods.

Answer

Thank-you, this has snow been clarified as follows:

“The eleven TTM therapists treated three patients each using the new 90 minute TTM protocol.”

Please see Lines 17 and 18 of the revised manuscript.

2. Comment

Line 21, 398. What does the values (2.89 to 1.76) mean? And I suggest that you provide the units.

Answer

The 95% of Confidence Interval (CI) of the mean difference between pre and post values have now been inserted as follows:

 “… (95% CI [1.76, 2.89 cm], p<0.001).”

The unit of measurement (i.e. “cm”) has also been added.

 See Lines 21, 22, 374 and 376.

3.Comment

Line 21, 22, 398, 400. If the values (2.33 and 0.37) are an average of 33 patients, please also show the SD.

Answer

The values refer to the CI and this has been clarified as follows:

“… a significant reduction in pain intensity of 2.33 cm (95% CI [1.76, 2.89 cm], p<0.001), and significant in-crease in Pain Pressure Threshold (PPT) of 0.37 kg/cm2 (95% CI [0.10, 0.64 kg/cm2], p<0.05)”

See Lines 21, 22, 548 and 550.

4. Comment

Line 22, 400. What does the values (0.10 to 0.64 kg/cm2 )

mean?

Answer

The valus refer to the CI and this has been clarified as follows:

“… (95% CI [0.10, 0.64 kg/cm2], p<0.05).”

See Line 550.

Introduction

1. Comment

Line 56-60. I suggest that you insert citations

Answer

Thank-you the relevant references are now cited:

 Please see [5,6] in Line 52.

2. Comment

Line 67-69. The meaning of the sentence is unclear, especially what does “feeling more alive” mean? I suggest that you explain that in a little more detail and/or insert citations.

Answer

As a result of shortening the Introduction as requested by Reviewer #2 this sentence has now been deleted.

3. Comment

Line 70-72. I suggest that you insert citations.

Answer

As a result of shortening the Introduction as requested by Reviewer #2 this sentence has now been deleted.

4. Comment

Line 81. Specifically I want you to explain what “physiological effect” is.

Answer

The study conducted by Plakornkul et al. (2016) investigated the physiological effect at Major Signal Points (MaSPs), specifically focusing on the increase in blood flow following massage treatment and the following clarification:

"especially the increase in blood flow."

has been added in Line 70.

5. Comment

Line 195. Is the sentence “11 TTM therapists, each with over 5 years of professional working experience, ...” true? On line 358, you described “the mean number of years of working experience was 4.1 ± 1.1 years ...”. This inconsistency can confuse the reader.

Answer

Thank-you and we apologize for the confusion. The sentence has been revised as follows:

" 11 TTM therapists, each with at least 2 years of professional working experience "

See Lines 152 and 153.

Materials and Methods

1. Comment

Line 333-337. When did you measure pain intensity? Immediately after the treatment? Also, in which body part did you measure pain intensity? Please be accurate.

Answer

VAS pain intensity and pressure pain threshold (PPT) were measured pre and post after a single treatment, for a central region in  Upper Trapezius muscle on the most painful side, and this has been clarified as follows:

 “… for the central part of Upper Trapezius muscle on the most painful side.”

See Lines 295 and 296.

Results

1. Comment

Line 363-365. I suggest that you provide the definition of “patients with OS” in this study and describe the symptoms of OS that the patients had in as much detail as possible. Based on your explanation,

Answer

Thank you more details about the definition of the patients with OS have been included as follows.

“The average BMI was 22.42 ± 3.51 kg/m2. Among the patients with OS, seven (21.1%) reported having underlying disease not requiring their exclusion from the study (e.g. allergy, stage 1 hypertension), an average number of hours worked per day of 7.85 ± 2.34 hours and an average number of days worked per week of 5.45 ± 0.87 days. The eighteen (54.4%) patients with OS had worked in their present occupation for an average of 12.21 ± 10.03 years (range 1 to 39 years). According to self-report using a VAS, pain intensity was reported as severe by 9 (27.3%) patients, as moderate by 18 (54.5%) patients, and as mild by 6 (18.2%) patients.

See Lines 324 and 329 to 335.

2. Comment

Line 374. It is unclear how the average was calculated and how the plots in Figure 5 were decided.

Answer

Thank you and we agree that further clarification is required. Accordingly, the following text has been added concerning the description of Figure 5:

“The confidence and satisfaction scores that were recorded for the individual TTM therapists and plotted in Figure 4 were averaged across all 11 therapists and the results are shown in Figure 5, where the average confidence scores are plotted above the average satisfaction scores. For each of Line, Hand and Force, the symbol is coloured red when the majority of the therapists gave a positive score, is an open circle when an equal number of therapists gave positive and negative scores, and is coloured blue when the majority of the therapists gave a negative score ”

See Lines 344 to 350.

3. Comment

Line 398-400. How did you measure pressure pain threshold? What kind of equipment was used in the measurement? Please be accurate in methods.

Answer

Thank-you we agree this is important and   the following sentence has been added:

“PPT was measured using a pressure algometry (OE-220, ITO Co., Ltd., Tokyo, Japan), for the central part of Upper Trapezius muscle on the most painful side.”

See Lines 294 to 296.

4. Comment

Line 404, 406. I suggest that you specify whether a, b, or c in Figure 6.

Answer

Thank you, Figure 6 for (a) and (b) have been deleted leaving only part Figure 6 (c) and which is now referred to as "Figure 6."

5. Comment

Figure 4 and 5. “not satisfied or not confident” was denoted by a blue symbol in Figure 4, whereas it was denoted by a green symbol in Figure 5. Unless there is a special reason, please unify the colors in the two figures for the avoidance of confusion.

Answer

Thank-you for identifying this inconsistency and Figure 5 has now been revised appropriately.

6. Comment

Figure 6. The sentence on line 397-400 and (a) and (b) of Figure 6 has the almost same content. I suggest that you reconsider whether you should include Figure (a) and (b).

Answer

Thank-you, Figure 6 for (a) and (b) have been deleted leaving only part Figure 6 (c) and which is now referred to as "Figure 6."

7. Comment

Figure 6. The meaning of arrows in Figure (a) and (b) is unclear. The meaning of the color difference (black, blue, and red) is also unclear. Please be accurate.

Answer

Thank-you, Figure 6 for (a) and (b) have been deleted leaving only part Figure 6 (c) and which is now referred to as "Figure 6."

Discussion

1. Comment

Line 438. I suggest that you interpret the finding of this study in a little more detail. For example, what are the possible mechanisms of significant improvement in the average values of PPT and VAS?

Answer

Thank-you we agree that the matter of interpretation is important and the following paragraph has been added to the Discussion;

“A possible mechanism for explaining the development of OS is that a combination of postural and psychological stress cause the stiffness of UT muscle to be increased, leading to functional limitations, impaired blood circulation and accumulation of pain-sensitizing substances [36], especially in MTrPs [37]. In OS patients MTrPs in are commonly found to lie along the length of the main axis of the anti-gravity muscles [34]. The 25 Step TTM protocol has been designed according to the concept of delivering a treatment that is of benefit to the whole-body, consisting of 20 Steps of acupressure massage, 2 occlusion Steps, and 3 steps of passive stretching. Several previous studies, have indicated that acupressure massage [38, 39], occlusion [17] and passive stretching [40] can all promote an increase in blood circulation, which may reduce the concentration of pain-sensitizing substances in stiff muscles [36,41] and result in a reduction in pain intensity and increase in PPT. Other possible mechanisms for the beneficial effect of TTM have been discussed by Keeratitanont et al. (2015) [16].”

See Lines 480 to 492.

2. Comment

Also, what is the possible reason that it was effective for some patients but not for others? This study included patients of different ages/genders. Is there any possibility that the treatment effects varied depending on ages/genders?

Answer

Thank-you we agree that this is an important consideration and the following paragraph has been added to the Discussion:

“Application of the new 25 Step 90 minute TTM protocol produced a reduction in pain intensity in 32 of the 33 patients with OS. For 16 of the patients pain intensity was reduced from severe or moderate pain to mild pain and in all these cases PPT was found to increase. However, for 4 of the 9 patients with severe pain PPT was observed to be decreased immediately after massage which may indicate that patients with severe pain should not be treated with TTM as they may exhibit possible adverse effects. The relatively small number of OS patients that were studied meant that is was not possible to perform sub-group analyseds to investigate potential effects of age and sex and which is an interesting topic for future research”

See Lines 471 to 479.

3. Comment

In addition, based on findings of this study, how can the new TTM be applied practically to clinical situations?

Answer

The following sentence is included within the response to Comment [2] above and which concerns the application of the new TTM protocol in clinical situations:

“However, for 4 of the 9 patients with severe pain PPT was observed to be decreased immediately after massage which may indicate that patients with severe pain should not be treated with TTM as they may exhibit possible adverse effects.”

See Lines 474 to 476.

Reviewer 2 Report

1. A reference to the arguments in line 38-42 seems necessary. Please add a reference.

2. There are too many statements without references. Please add references to every sentence if possible.

3. There are too many paragraphs in the introduction. Please shorten it to 4-5 paragraphs. Please organize only paragraphs related to the purpose of the study.

4. Please fill out the reliability and validity of the evaluation tool.

5.  Is there no normality test?

6.Further elucidation of the mechanism of how TTM affects patients with OS seems to be necessary. It is necessary to consider the overall effect of 25 steps and which step among them was more effective for OS patients.

7.Please check the spelling and dot(.) of the text.

8.Please add more clinical significance to the discussion

Author Response

Standardised 25 Step Traditional Thai Massage (TTM) Protocol for Treating Office Syndrome (OS) (ijerph-2288455)

We are very grateful to the two Reviewers and Editor for the very helpful reports they have kindly provided on the above manuscript that we have submitted for consideration for publication in the journal International Journal of Environmental Research and Public Health (IJERPH). The feedback is encouraging and we are pleased that the Reviewers consider the results that are presented to be clinically important. We have responded in full to all of the feedback that has been provided as is described below. All page numbers refers to the manuscript file with tracked changes.

Response to Reviewer #2

1. Comment

A reference to the arguments in line 38-42 seems necessary. Please add a reference.

Answer

We agree and have added the following reference and which is cited as [3].

 “Sucharit, W.; Eungpinichpong, W.; Hunsawong, T.; Pungsuwan, P.; Bennett, S.; Hojo, E.; Cruz, M.; Roberts, N.; Chatchawan, U. Pre-and Post-Treatment Study of the Application of a Traditional Thai Massage (TTM) Protocol for Treating Office Syndrome; article inpress.”

See Line 42.

2. Comment

There are too many statements without references. Please add references to every sentence if possible.

Answer

Thank-you we agree and have added the following citations of published papers:

[4] in Line 48

[5,6] in Line 52

[7,8] in Line 58

[9] in Line 61

[4] in Line 75

[14,15] in Line 87

[16] in Line 95

[4] in Line 105

[17] in Line 111

3. Comment

There are too many paragraphs in the introduction. Please shorten it to 4-5 paragraphs. Please organize only paragraphs related to the purpose of the study.

Answer

Thank-you, we agree and have removed four paragraphs so that the Introduction is shortened substantially.

4. Comment

Please fill out the reliability and validity of the evaluation tool.

Answer

Thank-you the following sentence has been added describing how repeatability and reproducibility of the PPT measurements were assessed|:

 “The intra-rater repeatability and inter-rater reproducibility of PPT measurements were tested in 28 volunteers before recruiting the participants. The results showed that the ICC was 0.94 (p < 0.001) and 0.97 (p < 0.001), respectively”.

See Lines 296 to 299.

5. Comment

Is there no normality test?

Answer

Yes normality of the data was assessed as described in the following sentence which has been added to the section describing the Statistical Analysis:

 “Normality of the outcome measures was investigated by using the Shapiro-Wilk Test.”

See Lines 309 to 310.

6. Comment

Further elucidation of the mechanism of how TTM affects patients with OS seems to be necessary. It is necessary to consider the overall effect of 25 steps and which step among them was more effective for OS patients.

Answer

This is a similar suggestion to Comment [1] concerning the Discussion by Reviewer #1 and has been addressed by addition of the following paragraph:

“The 25 Step TTM protocol has been designed according to the concept of delivering a treatment that is of benefit to the whole-body, consisting of 20 Steps of acupressure massage, 2 occlusion Steps, and 3 steps of passive stretching. Several previous studies, have indicated that acupressure massage [38, 39], occlusion [17] and passive stretching [40] can all promote an increase in blood circulation, which may reduce the concentration of pain-sensitizing substances in stiff muscles [36,41] and result in a reduction in pain intensity and increase in PPT. Other possible mechanisms for the beneficial effect of  TTM have been discussed by Keeratitanont et al. (2015) [16].”

See Lines 484 to 492.

7. Comment

Please check the spelling and dot(.) of the text.

Answer

Thank-you the manuscript has now been checked throughout for both spelling and punctuation.

8. Comment

Please add more clinical significance to the discussion

Answer

Thank-you the following sentence has been added to the Discussion concerning the clinical significance of the findings of the present study:

“Application of the new 25 Step 90 minute TTM protocol produced a reduction in pain intensity in 32 of the 33 patients with OS. For 16 of the patients pain intensity was reduced from severe or moderate pain to mild pain and in all these cases PPT was found to in-crease. However, for 4 of the 9 patients with severe pain PPT was observed to be decreased immediately after massage which may indicate that patients with severe pain should not be treated with TTM as they may exhibit possible adverse effects. The relatively small number of OS patients that were studied meant that is was not possible to perform sub-group analyseds to investigate potential effects of age and sex and which is an in-teresting topic for future research.”

See Lines 471 to 479.

Round 2

Reviewer 1 Report

I thank the authors for their satisfactory response to all my comments. I have no more comments.

Reviewer 2 Report

All corrected based on reviewer's comments.

Please check again according to the journal format.

Also, please check the grammar and spelling.